# Two novel genes identified by large-scale transcriptomic analysis are essential for biofilm and rugose colony development of *Vibrio vulnificus*

Hojun Lee[1,2], Hanhyeok Im[1,2], Seung-Ho Hwang[1,2¤a], Duhyun Ko[1,2¤b], Sang Ho Choi [1,2]*

**1** National Research Laboratory of Molecular Microbiology and Toxicology, Department of Agricultural Biotechnology, Seoul National University, Seoul, Republic of Korea, **2** Center for Food and Bioconvergence, and Research Institute of Agriculture and Life Science, Seoul National University, Seoul, Republic of Korea

¤a Current address: Microbiome Convergence Research Center, Korea Research Institute of Bioscience and Biotechnology, Daejeon, Republic of Korea
¤b Current address: Institute of Infectious Diseases, Seoul National University College of Medicine, Seoul, Republic of Korea
* choish@snu.ac.kr

**Data Availability Statement:** All relevant data are within the manuscript and its Supporting Information files.

## Abstract

Many pathogenic bacteria form biofilms to survive under environmental stresses and host immune defenses. Differential expression (DE) analysis of the genes in biofilm and planktonic cells under a single condition, however, has limitations to identify the genes essential for biofilm formation. Independent component analysis (ICA), a machine learning algorithm, was adopted to comprehensively identify the biofilm genes of *Vibrio vulnificus*, a fulminating human pathogen, in this study. ICA analyzed the large-scale transcriptome data of *V. vulnificus* cells under various biofilm and planktonic conditions and then identified a total of 72 sets of independently co-regulated genes, iModulons. Among the three iModulons specifically activated in biofilm cells, BrpT-iModulon mainly consisted of known genes of the regulon of BrpT, a transcriptional regulator controlling biofilm formation of *V. vulnificus*. Interestingly, the BrpT-iModulon additionally contained two novel genes, VV1_3061 and VV2_1694, designated as *cabH* and *brpN*, respectively. *cabH* and *brpN* were shared in other *Vibrio* species and not yet identified by DE analyses. Genetic and biochemical analyses revealed that *cabH* and *brpN* are directly up-regulated by BrpT. The deletion of *cabH* and *brpN* impaired the robust biofilm and rugose colony formation. CabH, structurally similar to the previously known calcium-binding matrix protein CabA, was essential for attachment to the surface. BrpN, carrying an acyltransferase-3 domain as observed in BrpL, played an important role in exopolysaccharide production. Altogether, ICA identified two novel genes, *cabH* and *brpN*, which are regulated by BrpT and essential for the development of robust biofilms and rugose colonies of *V. vulnificus*.

**Funding:** This work was supported by the National Research Foundation of Korea (NRF) and funded by the Ministry of Science, Information and Communications Technology, and Future Planning (https://www.msit.go.kr) (2021K1A3A1A20001134, to SHC). This work was also supported by Cooperative Research Program for Agriculture Science and Technology Development (Project No. PJ016298, to HL), Rural Development Administration (https://www.rda.go.kr/), Republic of Korea. The funders had no roles in study design, data collection and analysis, decision to publish, or preparation of the manuscript.

**Competing interests:** The authors have declared that no competing interests exist.

## Author summary

Biofilm formation is essential for pathogenic bacteria to survive under environmental stresses and host immune surveillances. Unlike conventional differential expression (DE) analysis, independent component analysis (ICA), a machine learning algorithm, could comprehensively analyze gene expression profiles under various conditions of biofilm and planktonic cells and identify the genes essential for biofilm formation. In this study, ICA decomposed large-scale transcriptome data of a devastating pathogen *Vibrio vulnificus* into the iModulons, sets of independently co-regulated genes. Among the iModulons activated in biofilm cells, BrpT-iModulon contained two novel genes not yet identified by DE analyses, *cabH* and *brpN*, in addition to the previously known genes of the BrpT regulon. Our results indicated that CabH and BrpN contribute to biofilm and rugose colony formation. CabH, carrying calcium-binding repeats, and BrpN, harboring an acyltransferase-3 domain, were identified to play a crucial role in surface attachment and exopolysaccharide production, respectively. Consequently, ICA newly identified *cabH* and *brpN* which are regulated by BrpT and are essential for robust biofilm and rugose colony development of *V. vulnificus*.

## Introduction

Bacteria often exist as a biofilm lifestyle, which is a sessile community attached to a surface [1]. Biofilms provide bacteria with protection from not only environmental stresses such as nutrient limitation but also host immune systems during infection [2]. Thus, biofilm formation is essential for the survival and virulence of a broad range of pathogens [3,4]. Biofilm formation includes sequential developmental stages composed of surface attachment, microcolony formation, and maturation into three-dimensional structures [1,5]. Mature biofilms are specialized and highly differentiated communities of bacteria covered by an extracellular polymeric matrix consisting of exopolysaccharides (EPSs), proteins, nucleic acids, and lipids [6]. Biofilm formation is essential for *Vibrio vulnificus*, a fulminating foodborne pathogen, to colonize and survive in oysters, the major infection route of the bacteria, and to cause diseases in mice [7–10].

Several studies have been conducted to understand the molecular mechanisms of biofilm formation in *V. vulnificus* [11–16]. BrpR, a transcriptional regulator governing biofilm formation in *V. vulnificus*, regulates downstream genes according to the intracellular levels of bis-(3′-5′)-cyclic dimeric guanosine monophosphate (c-di-GMP), which is a universal bacterial second messenger [11]. BrpR directly activates the expression of *brpLG* and *brpT* [11]. BrpL and BrpG play an important role in biofilm and rugose colony formation, through participating in EPS production [11]. BrpT is another transcriptional regulator controlling biofilm formation of *V. vulnificus* and activates the expression of *brpS*, *cabABC*, and the *brp* locus (*brpABCDFHIJK*) [12,13]. BrpS is also a transcriptional regulator that activates *cabABC* and represses *brpT* [13]. CabA is a calcium-binding protein and secreted to the cell exterior through CabB and CabC to form a structure of the biofilm matrix [14]. In concert with BrpL and BrpG, products of the *brp* locus also participate in the production of EPS [11,15]. In addition, another gene cluster named the *rbd* locus is involved in the biofilm formation of *V. vulnificus* [16]. The expression of the *rbd* locus is independent of c-di-GMP levels, and its activation signal has not yet been identified [16].

Differential expression (DE) analyses of the genes expressed more in a biofilm than in planktonic cells can simply predict a set of genes required for the biofilm formation under a

single condition [17–19]. However, diverse sets of genes are preferentially expressed in a bio-film depending on the conditions such as its developmental stages and growth environments [20,21]. Therefore, independent component analysis (ICA), a machine learning-based algorithm decomposing a mixture of signals into independent components [22,23], can be adopted to comprehensively identify the genes essential for biofilm formation under various conditions. ICA analyzes the genes expressed from the cells under various conditions and identifies the sets of independently co-regulated genes, iModulons [24–26]. Since the genes comprising a specific iModulon are co-regulated, the regulatory mechanism for the newly discovered genes in an iModulon can be easily predicted [27,28].

In this study, ICA decomposed the large-scale transcriptome data of *V. vulnificus* obtained from biofilm and planktonic cells under various conditions into the iModulons. Among the iModulons specifically activated in biofilm cells, BrpT-iModulon mainly consisted of known genes of the BrpT regulon. Interestingly, the BrpT-iModulon additionally included two novel genes, VV1_3061 and VV2_1694, designated as *cabH* and *brpN*, respectively. Transcript and molecular biological analyses demonstrated that BrpT positively regulates *cabH* and *brpN* by directly binding to their upstream regions. Genetic analyses showed that *cabH* and *brpN* contribute to the development of biofilms through enhanced surface attachment and EPS production, respectively. Taken together, ICA comprehensively analyzed the gene expressions under various conditions of the biofilm and planktonic cells of *V. vulnificus* and newly identified the genes of the BrpT regulon essential for biofilm formation.

## Results

### ICA identified 72 iModulons of *V. vulnificus*

Raw read counts of RNA-seq data from *V. vulnificus* cells under 14 biofilm and 147 planktonic, a total of 161 conditions were gathered to create large-scale transcriptome data (Fig 1A and S1 and S2 Datasets). ICA decomposed the transcriptome data into 83 independent components containing the entire genes with varied gene coefficients (Fig 1B and S3 Dataset). Simultaneously, ICA also calculated the activities of the independent components to account for the expression of the constituting genes under the various conditions (Fig 1B and S4 Dataset). The cumulative explained variance (CEV) of the 83 independent component activities covered 80% of the total explained variance of the large-scale transcriptome data calculated by principal component analysis (PCA) (S1A Fig), verifying that ICA appropriately decomposed the transcriptome data.

The varied gene coefficients represent the effect of the changed activities of the independent component on the expression levels of the constituting genes. A positive or negative gene coefficient implies that the increased activities of the independent component activated or repressed the constituting gene expression, respectively. Most of the gene coefficients are distributed around zero (S1B Fig), indicating that the alteration of the independent component activities significantly affects only a few genes. Thereby, the significantly affected genes in each independent component are defined as the element genes of a specific iModulon (Fig 1C, see Materials and Methods for details). After the removal of iModulons containing less than 4 element genes, a total of 72 iModulons were identified from the 83 independent components (S1 Table and S5 Dataset).

The resulting 72 iModulons were annotated with distinct biological functions based on the known or predicted roles of the element genes (S1 Table). Then, the iModulons were grouped into 10 categories, such as 'Structural components and signal transduction', 'Virulence and stress response', 'Translation', 'Metal homeostasis', 'Carbon and nitrogen source utilization', 'Amino acid and nucleotide biosynthesis', and 'Transport system', according to their biological

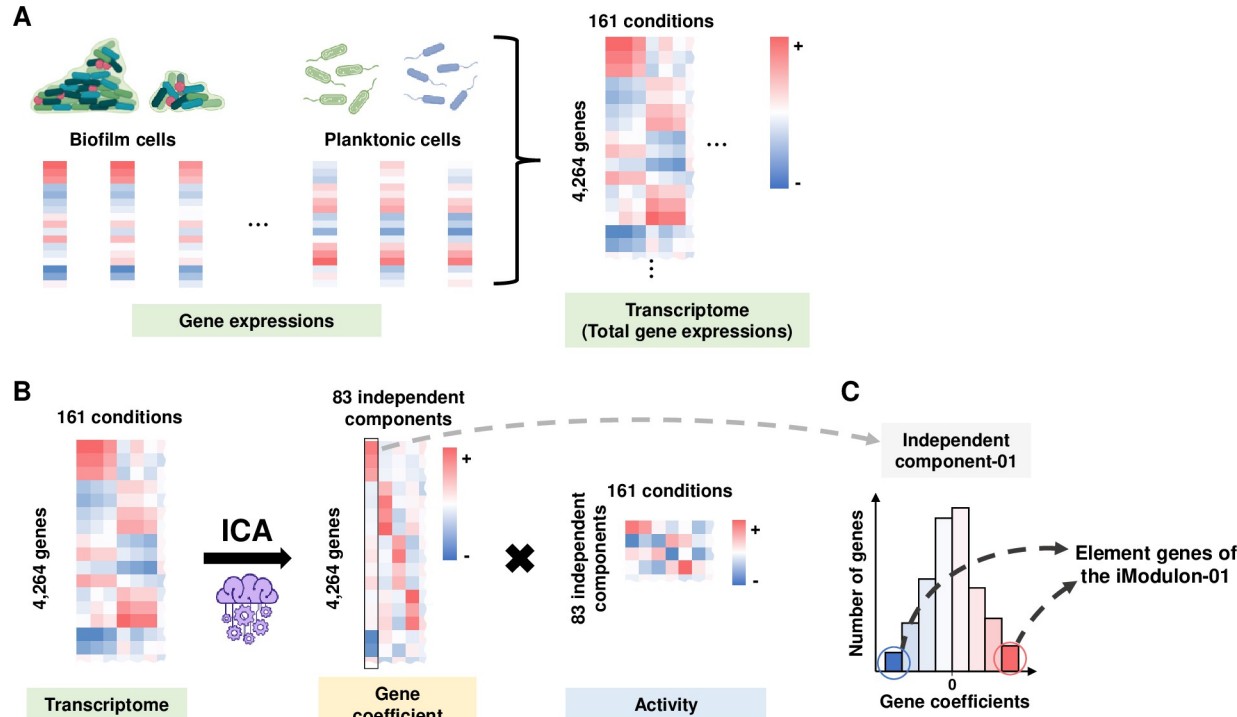

**Fig 1. Graphical illustration of the overall ICA workflow.** (A) Individual gene expression profiles from biofilm and planktonic cells under a total of 161 conditions were combined into the large-scale transcriptome data. Red and blue represent the high and low expression levels of the genes, respectively. (B) ICA decomposed the transcriptome data into 83 independent components and outputted two matrices, 'Gene coefficient' and 'Activity'. The 'Gene coefficient' matrix displayed the varied gene coefficients of 4,264 genes constituting the 83 independent components (S3 Dataset). The 'Activity' matrix indicated the activities of each independent component along the 161 conditions (S4 Dataset). (C) The independent component-01 was converted into the iModulon-01 by selecting the genes with significant gene coefficients. The same procedure was applied to the remaining 82 independent components.

functions (Fig 2). The iModulons grouped into the 'Genomic difference' category account for the difference in gene expression between strains, such as either *V. vulnificus* MO6-24/O and CMCP6 (S2A Fig), or wild-type and regulator knockout mutants (S2B Fig). The iModulons with biological functions that do not belong to other categories were grouped into the 'Miscellaneous functions' category. The iModulons whose biological functions are unclear were grouped into the 'Unknown' category.

## BrpT-iModulon contained *cabH* and *brpN*

To identify the iModulons containing the element genes essential for biofilm formation, the iModulon activities in the biofilm and planktonic cells were compared. The activities of iModulon-40, -57, and -64 significantly increased in the biofilm cells (S3 Fig), suggesting that their element genes are essential for biofilm formation. While most of the element genes comprising iModulon-40 and -64 were uncharacterized (S2 and S3 Tables), iModulon-57 consisted mostly of the genes of the regulon of BrpT, a known transcriptional regulator controlling biofilm formation of *V. vulnificus* (Fig 3A and Table 1) [11,13]. Also, the activities of iModulon-57 were greatly correlated with the expression levels of *brpT* (Pearson $R = 0.74$, $P < 10^{-10}$) (Fig 3B). These observations led us to designate the iModulon-57 as BrpT-iModulon. Interestingly, two hypothetical element genes, VV1_3061 and VV2_1694, which were not known to be regulated by BrpT, were also found in the BrpT-iModulon (Fig 3A and Table 1). Since the other element genes in the BrpT-iModulon were known to be regulated by BrpT, it was possible that

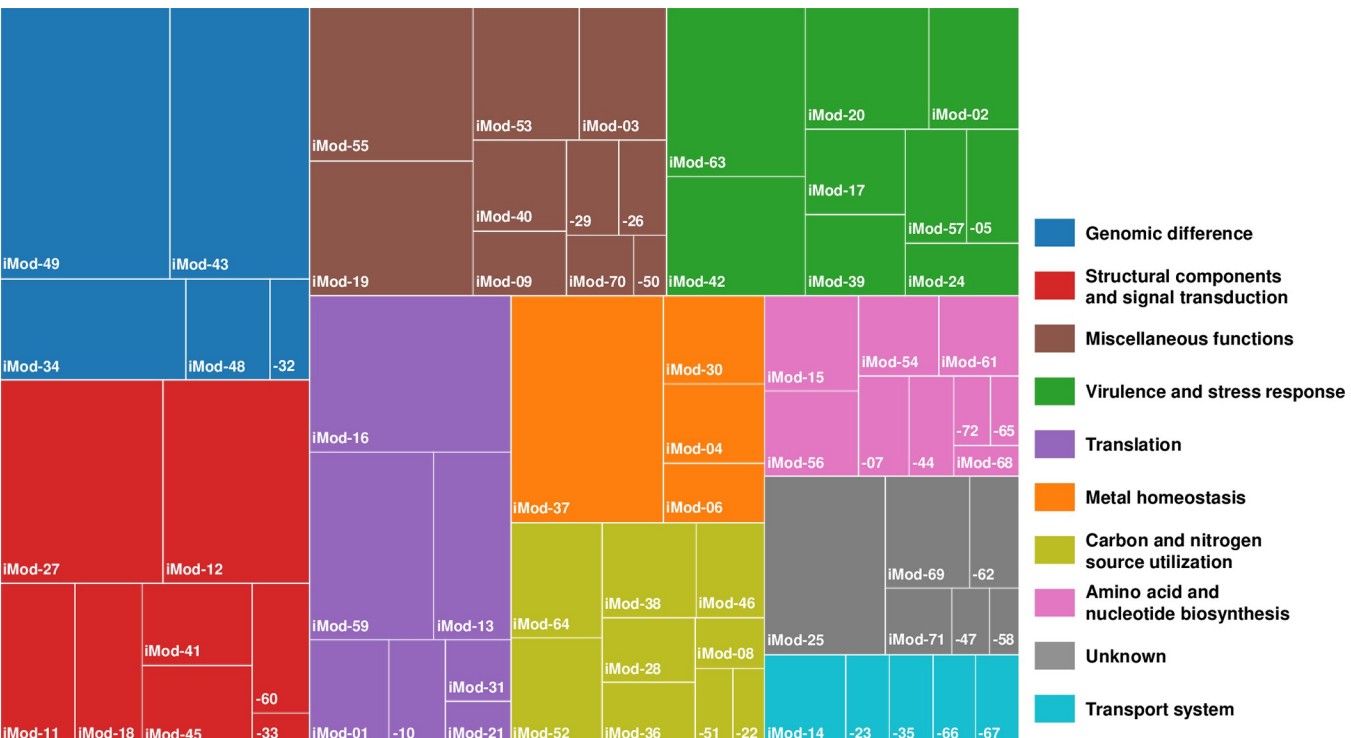

**Fig 2. Categorization of the iModulons by their biological functions.** 72 iModulons identified by ICA were grouped into 10 categories according to their biological functions as illustrated with different colors. The number of genes in each iModulon is indicated by the size of the box.

VV1_3061 and VV2_1694 are also the members of the BrpT regulon. To examine the possibility, the correlation between the expression levels of *brpT* and the VV1_3061 or VV2_1694 genes was examined. As shown in Fig 3C and 3D, the expression of VV1_3061 and VV2_1694 was positively correlated with that of *brpT* (Pearson $R = 0.67$, $P < 10^{-10}$ and Pearson $R = 0.46$, $P < 10^{-9}$, respectively), proposing that VV1_3061 and VV2_1694 might be activated by BrpT.

The open reading frames (ORFs) of VV1_3061 and VV2_1694 were predicted to encode a calcium-binding protein and an acyltransferase, respectively (Table 1). In *V. vulnificus*, *cabA* and *brpL*, the essential biofilm genes, also encode a calcium-binding protein and an acyltransferase, respectively [11, 14]. Thus, the structural similarities between the proteins encoded by VV1_3061 and *cabA*, and VV2_1694 and *brpL* were further investigated. As shown in Fig 4A, the repeats in the protein encoded by VV1_3061 were comparable to the calcium-binding repeats of CabA. Moreover, the structural superimposition of the two proteins showed the root-mean-square deviation (RMSD) value of 0.550 Å, indicating that these are highly analogous (Fig 4B). The structural similarity of the proteins led us to designate VV1_3061 as *cabH*. Meanwhile, a domain of the protein encoded by VV2_1694 was homologous to the acyltransferase-3 (AT3) domain of BrpL (Fig 4A). Although the whole structures of the proteins were not highly analogous (Fig 4C), the domain homologous to the AT3 domain of BrpL led us to designate VV2_1694 as *brpN*.

## *cabH* and *brpN* are positively and directly regulated by BrpT

To experimentally verify that *cabH* and *brpN* are positively regulated by BrpT, the expression of *cabH* and *brpN* was compared in the *brpT*-deletion mutant (Δ*brpT*) and its isogenic parent

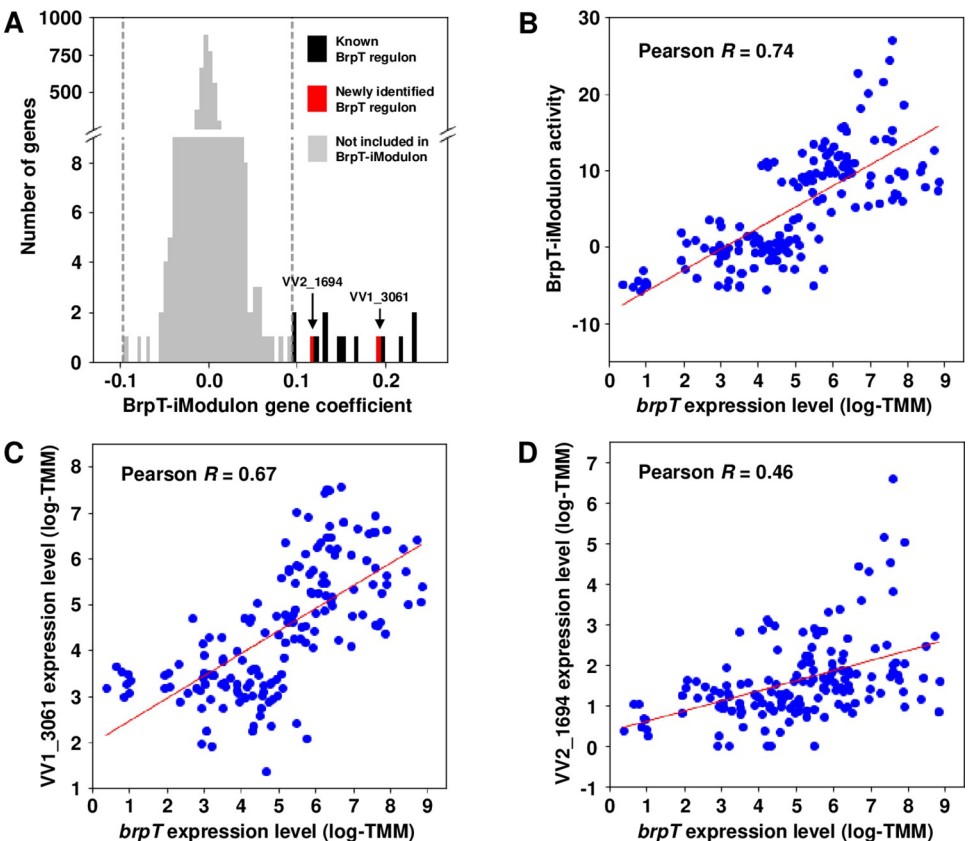

**Fig 3. Validation of the BrpT-iModulon.** (A) A histogram of the gene coefficients in the BrpT-iModulon. The gray dashed lines show the cutoff values of the gene coefficients. Newly identified members of the BrpT regulon, VV1_3061 and VV2_1694, are presented with red bars. (B to D) Scatter plots of the expression levels of *brpT* and the activities of the BrpT-iModulon (B), the expression levels of VV1_3061 (C) and VV2_1694 (D). The expression levels are represented as log-TMM, log-transformed read counts after normalization using the trimmed mean of M-values (TMM) method. The Pearson *R* values between the two variables are denoted in the boxes. Each dot of the plots represents a single biological replicate. The red lines represent each regression line of the plots.

strain. Since BrpT is involved in c-di-GMP-induced biofilm formation [13,15], the JN111 strain, whose intracellular c-di-GMP levels are elevated by the addition of arabinose [14], was used as the parent strain in this study. The transcript levels of *cabH* and *brpN* in Δ*brpT* were significantly decreased compared with those in the parent strain (Fig 5A). The decreased transcript levels of *cabH* and *brpN* in Δ*brpT* were restored by complementation of *brpT* (Fig 5B). These results confirmed that the expression of *cabH* and *brpN* is positively regulated by BrpT. To investigate whether BrpT regulates *cabH* and *brpN* directly, BrpT was purified, and its binding to the upstream regions of *cabH* and *brpN* was examined by electrophoretic mobility shift assays (EMSAs). The addition of BrpT to the labeled DNA probes resulted in retarded bands in a BrpT concentration-dependent manner (Fig 5C and 5D). The binding of BrpT was specific because the assays were performed in the presence of poly(dI-dC) as a non-specific competitor. Moreover, the same unlabeled DNA fragment, which was used as a self-competitor, showed competition for the BrpT binding in a dose-dependent manner (Fig 5C and 5D), verifying the specific binding of BrpT to the upstream regions of *cabH* and *brpN*. Taken together, these results indicated that BrpT regulates the expression of *cabH* and *brpN* by directly binding to their upstream regions.

**Table 1. The element genes included in the BrpT-iModulon.**

| Locus tag[a] | Gene[b] | Gene coefficient[c] | Annotation[d] |
|---|---|---|---|
| VV1_3061 | - | 0.1905 | calcium-binding protein |
| VV2_1569 | *brpS* | 0.0962 | LuxR C-terminal-related transcriptional regulator |
| VV2_1571 | *cabA* | 0.1459 | biofilm matrix calcium-binding repeat protein CabA |
| VV2_1572 | *cabB* | 0.1235 | type I secretion system permease/ATPase |
| VV2_1573 | *cabC* | 0.0957 | HlyD family type I secretion periplasmic adaptor subunit |
| VV2_1575 | *brpJ* | 0.1332 | oligosaccharide flippase family protein |
| VV2_1576 | *brpI* | 0.1989 | glycosyltransferase |
| VV2_1577 | *brpH* | 0.2311 | putative capsular polysaccharide synthesis family protein |
| VV2_1578 | *brpF* | 0.1316 | glycosyltransferase |
| VV2_1579 | *brpD* | 0.1674 | polysaccharide biosynthesis tyrosine autokinase |
| VV2_1580 | *brpC* | 0.1545 | polysaccharide export protein |
| VV2_1581 | *brpB* | 0.2184 | outer membrane beta-barrel protein |
| VV2_1582 | *brpA* | 0.2314 | undecaprenyl-phosphate glucose phosphotransferase |
| VV2_1694 | - | 0.1159 | acyltransferase |

[a, b, and d] Locus tags, gene names, and annotations are based on the *V. vulnificus* CMCP6 genome (GenBank accession numbers: AE016795.3 and AE016796.2).

[b] Unnamed genes are described as '-'.

[c] Gene coefficients of the element genes in the iModulon are based on S3 Dataset.

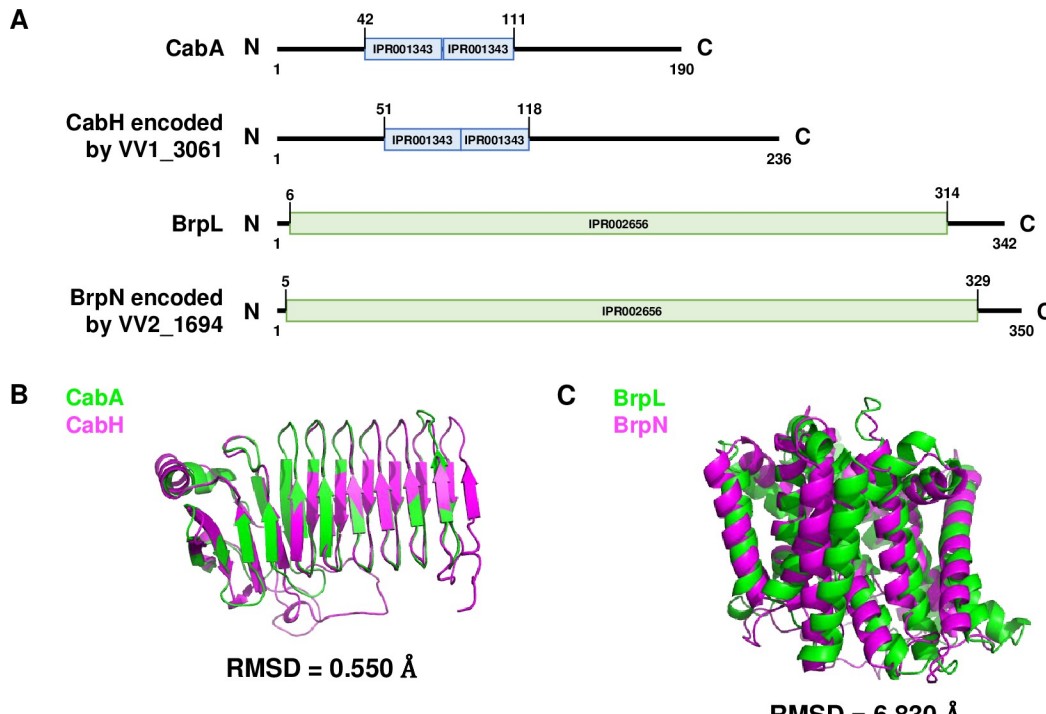

**Fig 4. Predicted protein domains and structures of CabA, CabH, BrpL, and BrpN.** (A) Protein repeats and domains of CabA, CabH, BrpL, and BrpN predicted by using InterPro. The numerical values represent the amino acid numbers of each protein. IPR001343, RTX calcium-binding nonapeptide repeat; IPR002656, acyltransferase-3 domain. (B, C) Superimpositions of the predicted protein structures of CabA (green) and CabH (magenta) (B), and BrpL (green) and BrpN (magenta) (C). Protein structures were predicted by using the Alphafold2 algorithm and the superimpositions of the proteins were performed by using PyMOL. RMSD, root-mean-square deviation.

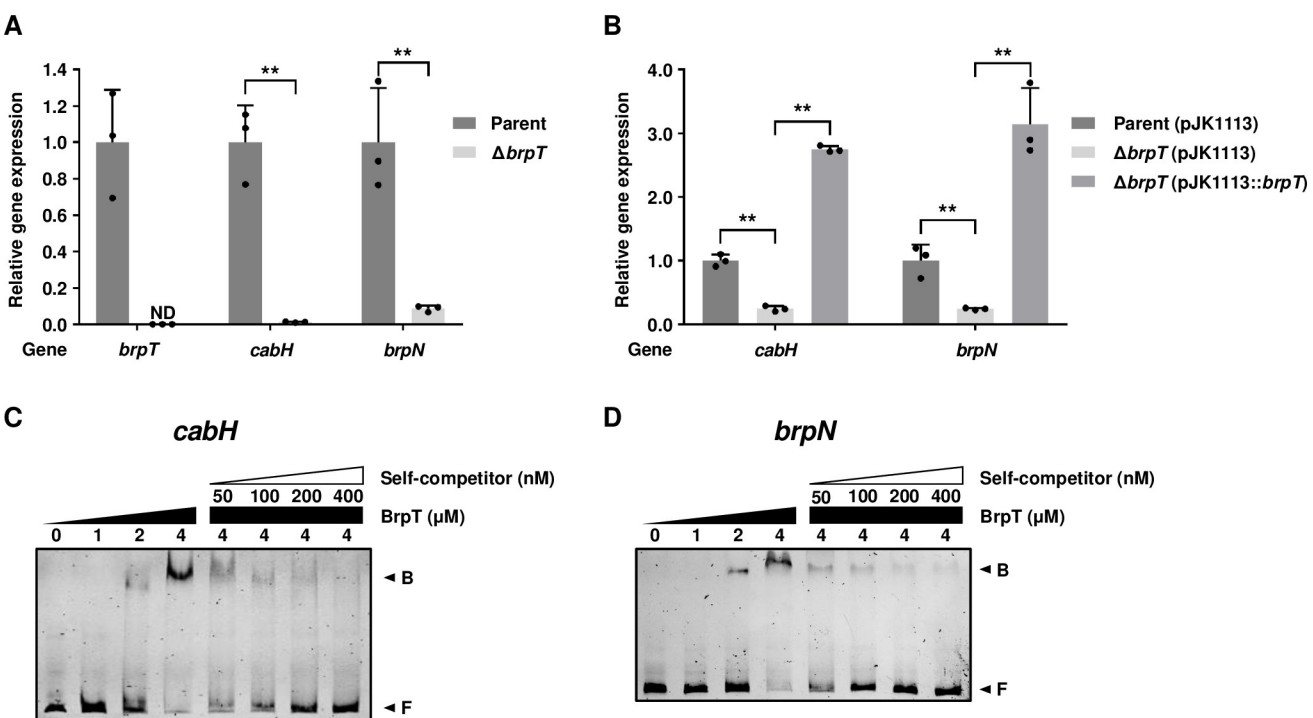

**Fig 5. Direct regulation of the *cabH* and *brpN* expression by BrpT.** (A, B) Effects of BrpT on the expression of *cabH* and *brpN*. Total RNAs were isolated from the *V. vulnificus* strains grown in LB supplemented with 0.01% (w/v) arabinose (A) or 0.03% arabinose, 100 µg/ml kanamycin, and 100 µg/ml ampicillin for complementation experiments (B). The expression of *brpT*, *cabH*, and *brpN* was determined by qRT-PCR analysis, and the expression of each gene in the parent strain was set as 1 (*n* = 3). Error bars represent the SD. Statistical significance was determined by the Student's *t* test (\*\*, *P* < 0.005). ND, not detected. Parent and Parent (pJK1113), parent strain; Δ*brpT* and Δ*brpT* (pJK1113), *brpT* mutant; Δ*brpT* (pJK1113::*brpT*), complemented strain. (C, D) Direct binding of BrpT to upstream regions of *cabH* and *brpN*. The 6-FAM labeled DNA fragments (10 nM) of the upstream regions of *cabH* (C) and *brpN* (D) were incubated with increasing amounts of BrpT as indicated. For competition analysis, the same but unlabeled DNA fragments were used as self-competitors. Various amounts of self-competitors were added as indicated to the reaction mixtures before the addition of BrpT (4 µM). Each gel representing the mean result from at least three independent experiments was photographed using ChemiDoc Touch Imaging System (Bio-Rad). B, bound DNA; F, free DNA.

## CabH and BrpN are essential for biofilm and rugose colony formation

Since *cabH* and *brpN* were included in the BrpT-iModulon that was activated in biofilm cells (Table 1 and S3B Fig), the involvement of *cabH* and *brpN* in biofilm formation was further investigated. The biofilm levels of the *cabH*-deletion mutant (Δ*cabH*) and *brpN*-deletion mutant (Δ*brpN*) were reduced when compared with those of the parent strain (Fig 6A). The reduced biofilm levels of Δ*cabH* and Δ*brpN* were restored by complementation of *cabH* and *brpN*, respectively (Fig 6B). These results indicated that CabH and BrpN are crucial for biofilm formation of *V. vulnificus*. In addition, the biofilm level of Δ*brpT* decreased when compared with that of the parent strain (Fig 6A), further supporting the previous observation that *cabH* and *brpN* are activated by BrpT (Fig 5).

The colony morphology of Δ*cabH* and Δ*brpN* was also compared with that of the parent strain. In accordance with the diminished biofilm formation (Fig 6), Δ*cabH* and Δ*brpN* exhibited reduced colony rugosity when compared with the parent strain (Fig 7A). The changed colony morphologies of Δ*cabH* and Δ*brpN* were recovered by complementation of *cabH* and *brpN*, respectively (Fig 7B). These results demonstrated that CabH and BrpN are important for the rugose colony development of *V. vulnificus*. Furthermore, the reduction of colony rugosity caused by the deletion of *brpN* was more severe than that of *cabH* (Fig 7A and 7B), suggesting that BrpN plays a more crucial role in rugose colony development. Additionally, Δ*brpT*

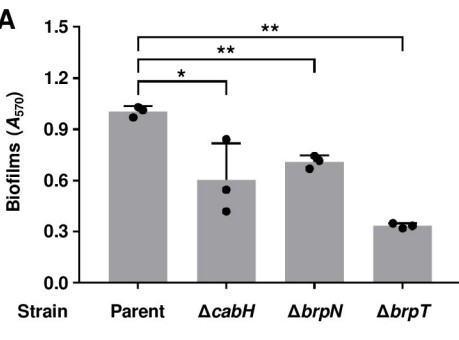
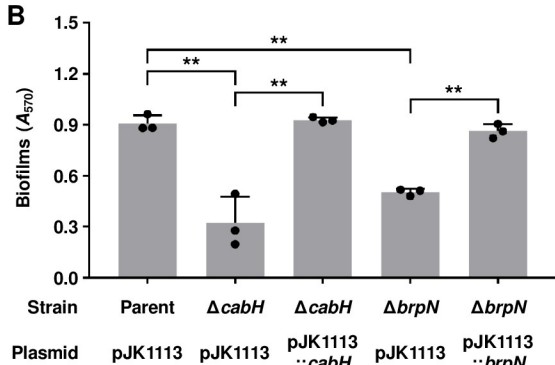

**Fig 6. Biofilm formation of the *cabH* and *brpN* mutants.** (A, B) For quantitative analysis of the biofilm, biofilms of the *V. vulnificus* strains were grown in VFMG supplemented with 0.01% arabinose (A) or 0.01% arabinose, 100 μg/ml kanamycin, and 100 μg/ml ampicillin for complementation experiments (B) in 96-well polystyrene microtiter plates for 24 h. Then supernatants were removed from the wells and the remaining biofilms were stained with 1% crystal violet. The crystal violet was eluted and its $A_{570}$ was determined to quantify the biofilms ($n = 3$). Error bars represent the SD. Statistical significance was determined by the Student's *t* test (*, $P < 0.05$; **, $P < 0.005$). Parent and Parent (pJK1113), parent strain; Δ*cabH* and Δ*cabH* (pJK1113), *cabH* mutant; Δ*brpN* and Δ*brpN* (pJK1113), *brpN* mutant; Δ*brpT*, *brpT* mutant; Δ*cabH* (pJK1113::*cabH*) and Δ*brpN* (pJK1113::*brpN*), complemented strains.

showed a smooth colony (Fig 7A), again supporting the previous observation that *cabH* and *brpN* are activated by BrpT (Fig 5).

## CabH is crucial for surface attachment

Biofilms of Δ*cabH* and Δ*brpN* were further visualized in large scales using polystyrene and glass test tubes. As shown in Fig 8, when compared with the parent strain, Δ*cabH* exhibited reduced biofilm formation, especially defects in the attachment to both polystyrene and glass tubes. These observations suggest that CabH contributes to robust biofilm formation by enhancing surface attachment. The role of CabH on surface attachment was further compared with that of CabA, a calcium-binding protein [14]. As shown in Fig 8, the *cabA*-deletion mutant (Δ*cabA*) reduced biofilm formation of *V. vulnificus*, but the attachment of Δ*cabA* to the test tubes was comparable with that of the parent strain, confirming the previous observation that CabA is important for biofilm structural development rather than surface attachment [14]. Additionally, one interesting point was that among the mutants, Δ*cabH* specifically showed more reduced biofilm formation in smooth-surfaced glass tubes than in polystyrene tubes (Fig 8). The result suggests that Δ*cabH* with impaired surface attachment has more difficulties in attaching to smoother surfaces.

## BrpN is involved in EPS production

Previously, BrpN was predicted as an AT3 domain-containing acyltransferase (Fig 4A) and essential for the biofilm and rugose colony formation (Figs 6, 7, and 8). However, the attachment of Δ*brpN* to the test tubes was comparable with that of the parent strain (Fig 8A and 8B), suggesting that BrpN contributes to biofilm and rugose colony formation through a function other than surface attachment. Since the AT3 domain-containing acyltransferase is known to contribute to EPS biosynthesis [29, 30], the EPS production of Δ*brpN* was further compared with that of the parent strain using SDS-PAGE. EPS extract from the parent strain showed a high intensity, but that from Δ*brpN* exhibited a much lower intensity (Fig 9). This result confirmed that BrpN participates in EPS production, implying that reduced biofilm and rugose colony formation of Δ*brpN* (Figs 6, 7, and 8) is due to diminished EPS production. In addition,

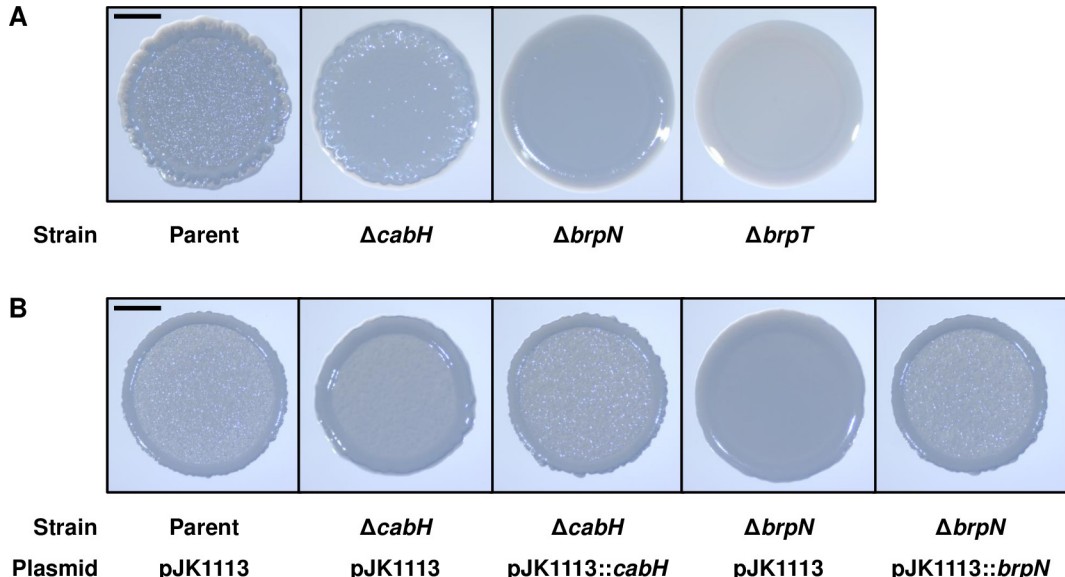

**Fig 7. Colony morphology of the *cabH* and *brpN* mutants.** (A, B) The *V. vulnificus* strains were spotted onto VFMG agar supplemented with 0.02% arabinose (A) or 0.02% arabinose, 100 μg/ml kanamycin, and 100 μg/ml ampicillin for complementation experiments (B) and incubated for 24 h. Each colony representing the mean rugosity from at least three independent experiments was visualized using a stereomicroscope (Stemi 305, Zeiss). All images are shown at the same scale, and 1-mm scale bars are shown on the images of the parent strain. Parent and Parent (pJK1113), parent strain; Δ*cabH* and Δ*cabH* (pJK1113), *cabH* mutant; Δ*brpN* and Δ*brpN* (pJK1113), *brpN* mutant; Δ*brpT*, *brpT* mutant; Δ*cabH* (pJK1113::*cabH*) and Δ*brpN* (pJK1113::*brpN*), complemented strains.

the role of BrpN in EPS production was further compared with that of another AT3 domain-containing protein BrpL. As shown in Fig 9, the EPS production from Δ*brpN* was lower than that from the *brpL*-deletion mutant (Δ*brpL*). Furthermore, the EPS production from the *brpN brpL* double-deletion mutant (Δ*brpN* Δ*brpL*) was even lower than that from either Δ*brpN* or Δ*brpL* (Fig 9). Together, these results suggest that BrpN is more crucial than BrpL in EPS production, and their activities for EPS production are additive.

## Discussion

In this study, ICA decomposed the large-scale transcriptome data from 14 biofilm and 147 planktonic cells of *V. vulnificus* into 72 iModulons (Figs 1 and 2, S1 Table, and S1, S2, and S5 Datasets). Among the iModulons, iModulon-40, -57, and -64 showed significantly increased activities in the biofilm cells (S3 Fig). iModulon-40 included the *rbdIJ* genes, which contribute to biofilm formation (S2 Table and S4A Fig) [16]. The signal for the expression of *rbdIJ* is not c-di-GMP, a known signal for bacterial biofilm formation, and rather still unknown [16]. Therefore, identification of the actual signal for the expression of *rbdIJ* could extend our understanding of the mechanisms for *V. vulnificus* biofilm formation. Meanwhile, most of the characterized element genes in iModulon-64 were involved in central carbon metabolism (S3 Table and S4B Fig). Biofilm formation can be affected by the availability of carbon sources, and thereby is regulated concurrently with central carbon metabolism in several pathogens [31–33]. Further investigation of the regulatory networks of iModulon-64 may reveal the link between biofilm formation and the nutritional availability of *V. vulnificus*.

iModulon-57, designated as BrpT-iModulon, contained two hypothetical element genes, VV1_3061 and VV2_1694, which are located apart from the previously known BrpT regulon and thus have not yet been recognized as the BrpT regulon (Fig 3 and Table 1). The product of

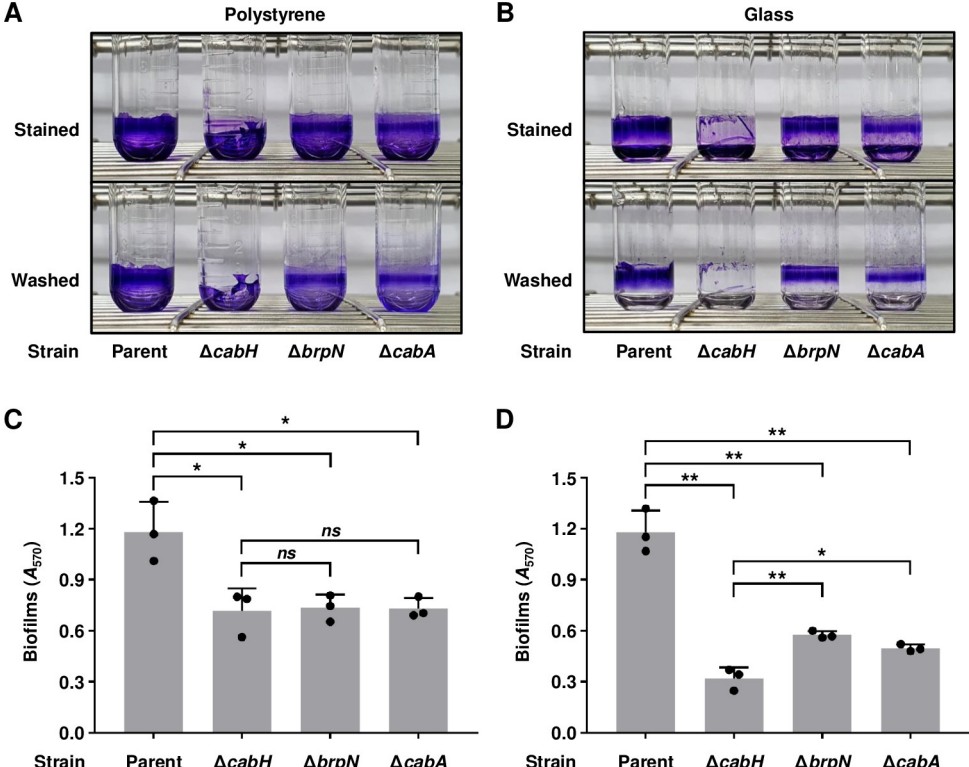

**Fig 8. Surface attachment and biofilm formation of the *cabH*, *brpN*, and *cabA* mutants.** (A, B) Biofilms of the *V. vulnificus* strains were grown in VFMG supplemented with 0.01% arabinose in polystyrene (A) or glass (B) test tubes for 24 h. Then supernatants were removed from the tubes and the remaining biofilms were stained with 1% crystal violet. The stained biofilms were then washed with vibration in PBS to remove loosely attached cells. To evaluate the attachment to the test tube surface, each test tube representing the mean result from at least three independent experiments was photographed using a mobile camera. Stained, photographed after the removal of the crystal violet solution; Washed, photographed after being washed with PBS. (C, D) After being washed with PBS, biofilms in the polystyrene (C) or glass (D) test tubes were quantified by eluting the crystal violet and measuring its $A_{570}$ ($n = 3$). Error bars represent the SD. Statistical significance was determined by the Student's *t* test ($^*$, $P < 0.05$; $^{**}$, $P < 0.005$; *ns*, not significant). Parent, parent strain; Δ*cabH*, *cabH* mutant; Δ*brpN*, *brpN* mutant; Δ*cabA*, *cabA* mutant.

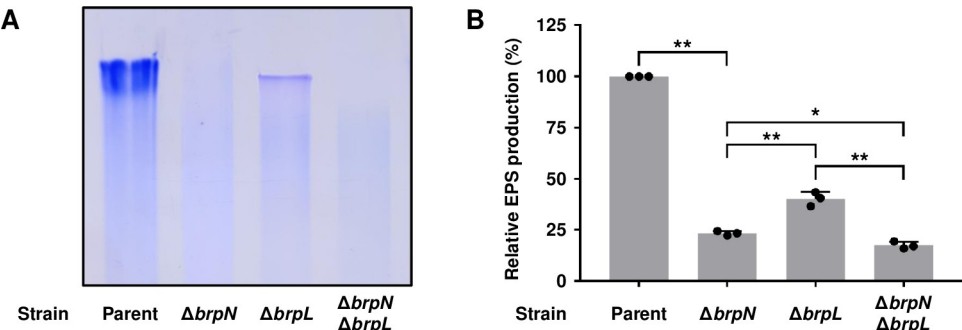

**Fig 9. EPS production of the *brpN*, *brpL*, and *brpN brpL* double mutants.** (A) EPS extracts were prepared from the *V. vulnificus* strains and then resolved on a 4% polyacrylamide gel by SDS-PAGE. The gel was stained with Stains-All and the gel representing the mean result from at least three independent experiments was photographed using a mobile camera. (B) EPS production of the mutants was quantified from the intensity of each lane, and the production of the parent strain was set as 100% in each experiment ($n = 3$). Error bars represent the SD. Statistical significance was determined by the Student's *t* test ($^*$, $P < 0.05$; $^{**}$, $P < 0.005$). Parent, parent strain; Δ*brpN*, *brpN* mutant; Δ*brpL*, *brpL* mutant; Δ*brpN* Δ*brpL*, *brpN brpL* double mutant.

VV1_3061 was named CabH and VV2_1694 was named BrpN, based on the structural similarity with CabA and domain homogeneity with BrpL, respectively (Fig 4). The deletion of *brpT* significantly decreased the expression levels of *cabH* and *brpN* when compared with those in the parent strain (Fig 5A and 5B). In addition, the direct and specific binding of BrpT to the upstream regions of *cabH* and *brpN* was demonstrated (Fig 5C and 5D). The combined results indicated that *cabH* and *brpN* are the novel members of the BrpT regulon and up-regulated directly by BrpT. Interestingly, *cabH* and *brpN* have not yet been identified by comparisons of the biofilm and planktonic transcriptome data of *V. vulnificus* under a specific condition (S5 Fig). These results suggest that ICA decomposition of the large-scale transcriptome data under various biofilm and planktonic conditions can uncover novel genes involved in bacterial biofilm formation, yet unrevealed by conventional DE analyses.

The *cabH* gene is a newly identified member of the BrpT regulon and predicted to encode CabH containing calcium-binding repeats (Fig 4A). Homologs of CabH are found in other *Vibrio* species such as *Vibrio* sp. 03_296, *V. fluvialis*, and *V. parahaemolyticus*, indicating that the protein may be conserved in other *Vibrio* species. However, no functional roles of CabH have yet been identified. Deletion of *cabH* reduced biofilm levels along with the decreased attachment of *V. vulnificus* to the test tubes (Figs 6 and 8), suggesting that CabH contributes to biofilm formation by playing a crucial role in surface attachment. The protein structure of CabH is highly analogous to CabA (Fig 4B), which is secreted to the cell exterior by the CabBC secretion system and a component of a biofilm matrix [14]. In addition, *cabH* is activated by BrpT along with *cabBC* (Fig 5) [13]. These findings suggest that CabH is also a biofilm matrix component secreted by CabBC. The reduced colony rugosity of Δ*cabH* further suggests that CabH is a biofilm matrix constituent secreted out of the cell (Fig 7). Moreover, since CabH possesses calcium-binding repeats similar to those of CabA (Fig 4A), CabH would be functional when the extracellular calcium is present, as observed in CabA [14].

The *brpN* gene is another newly identified member of the BrpT regulon and predicted to encode an acyltransferase containing an AT3 domain (Fig 4A), which is involved in EPS biosynthesis [29,30]. Homologs of BrpN are found in *Vibrio* species such as *Vibrio* sp. 03_296 and *V. mytili*, indicating that the protein is also conserved in other *Vibrio* species. However, no functional roles of BrpN have yet been identified. Deletion of *brpN* reduced biofilm formation and colony rugosity along with decreased EPS production of *V. vulnificus*, suggesting that BrpN contributes to biofilm and rugose colony development by producing EPS (Figs 6, 7, 8, and 9). Additionally, EPS production was more diminished in Δ*brpN* than in Δ*brpL*, and further reduced in Δ*brpN* Δ*brpL* (Fig 9). Consequently, the biofilm level and colony rugosity were more decreased in Δ*brpN* than in Δ*brpL*, and further diminished in Δ*brpN* Δ*brpL* (S6 Fig). These results suggest that BrpN is more crucial for EPS production, biofilm formation, and rugose colony development than BrpL, and the two acyltransferases function additively. Interestingly, *brpN* is in chromosome II of *V. vulnificus*, distantly located from the other EPS genes involving *brpLG* and the *brp* locus. Moreover, the GC content of *brpN* is significantly different from that of the other EPS genes (S4 Table), indicating that *V. vulnificus* may have received *brpN* transferred horizontally to further elaborate its EPS production [34,35].

As depicted in Fig 10, this study expanded the current understanding of *V. vulnificus* biofilm formation using ICA. ICA decomposed the large-scale transcriptome data from the biofilm and planktonic cells under diverse conditions into iModulons. Further investigation of the BrpT-iModulon, activated in the biofilm cells, discovered *cabH* and *brpN* as the new members of the BrpT regulon. BrpT directly activates the expression of *cabH* and *brpN* in addition to *brpS*, the *cabABC* operon, and the *brp* locus. BrpS activates *cabABC* expression and represses *brpT* expression to constitute a negative feedback loop tuning the *brpT* expression precisely. CabH, structurally similar to a biofilm matrix protein CabA, has calcium-binding

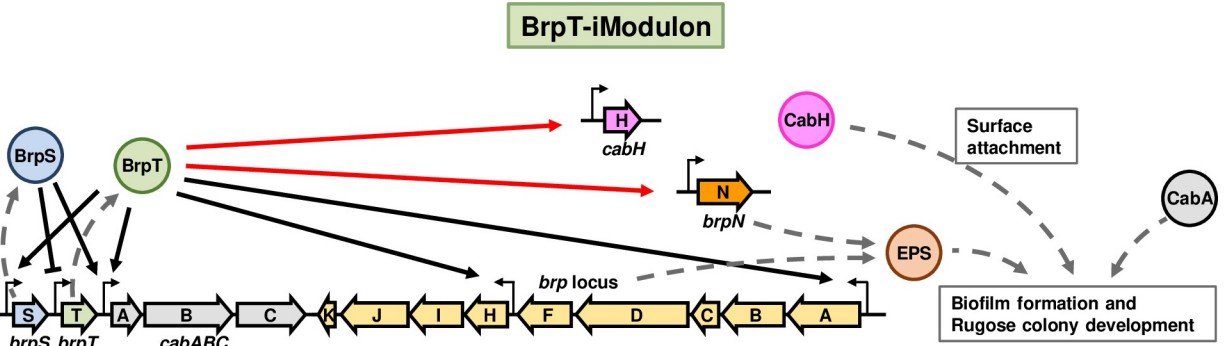

**Fig 10. The transcriptional networks of BrpT.** BrpT-iModulon contained *cabH* and *brpN*, the newly discovered genes in this study, in addition to *brpS*, *cabABC*, and the *brp* locus. Transcriptional regulator BrpT activates the expression of *cabH*, *brpN*, *brpS*, *cabABC*, and the *brp* locus. Transcriptional regulator BrpS activates the expression of *cabABC* and represses the expression of *brpT*. CabH is involved in the surface attachment and CabA is crucial for biofilm structural development. BrpN and the proteins expressed from the *brp* locus participate in the EPS biosynthesis. This sophisticated regulatory network elaborately controls robust biofilm and rugose colony development in *V. vulnificus*.

repeats and contributes to the surface attachment. BrpN carries an AT3 domain and participates in the biosynthesis of EPS along with the proteins encoded by the *brp* locus. Consequently, ICA identified the novel genes of the BrpT regulon, *cabH* and *brpN*, which were not discovered by conventional DE analyses, but essential for robust biofilm and rugose colony development of *V. vulnificus*. Accordingly, ICA can also be adopted to identify the genes required for biofilm formation of other pathogens and to understand their characteristics.

## Materials and methods

### Generation of the input data set of ICA comprising the *V. vulnificus* transcriptome data

Raw sequencing reads of available RNA-seq data of *V. vulnificus* were collected from the Sequence Read Archive (SRA) database at NCBI (https://www.ncbi.nlm.nih.gov/sra) (S1 Dataset). All reads were trimmed using Trimmomatic (v0.39) [36] with the following options "LEADING:3 TRAILING:3 SLIDINGWINDOW:4:15 MINLEN:36". The trimmed reads were mapped to the reference genome of *V. vulnificus* (GenBank accession numbers NC_014965.1 for chromosome I and NC_014966.1 for chromosome II) using STAR aligner (v2.7.8a) with the following options "—outSAMtype BAM—alignIntronMax 1" [37]. The mapped reads were counted using HTSeq (v0.13.5) with the following options "-s no -r pos -t gene -f bam -m intersection-strict" [38]. To ensure the quality of the data, the genes with less than 10 counts per million (CPM) were removed before further analysis. The read counts were normalized using the trimmed mean of M-values (TMM) method from the *edgeR* package and log-transformed by $\log_2(TMM+1)$, referred to as log-TMM [39,40]. Biological replicates with $R^2 < 0.9$ between log-TMM values were removed to eliminate the technical noise, and the remaining transcriptome data were used as the input data set of ICA (S2 Dataset).

### Identification of iModulons using ICA

ICA was performed on the input data set as described previously [25,28]. The FastICA algorithm from the Scikit-learn Python package (v0.22.1) was executed 256 times with random seeds to construct independent components [41,42]. The resulting independent components were clustered using the DBSCAN algorithm from the Scikit-learn Python package to identify the reproducible components [43]. To select the co-regulated genes of each independent

component, the gene with the largest absolute gene coefficient value was removed repeatedly, and D'Agostino's $K^2$ test statistic was calculated for each remaining distribution after the removal [44]. Once the test statistic dropped below a cutoff, the removed genes were designated as the co-regulated genes, the element genes of the iModulons.

To decide the $K^2$ test statistic cutoff, RegulonDB (v10.9) was used to perform a two-sided Fisher's exact test (FDR < $10^{-5}$) between the top 30 element genes with the highest absolute gene coefficients of each independent component and the previously known genes of the regulon of every regulator [45,46]. Among the regulators, a regulator with the lowest *P* value was linked to each independent component. Then, the F1-scores were calculated between the regulon of the independent component-linked regulators and the element genes in the independent components selected on the basis of the $K^2$ test statistic cutoff varying from 200 to 1,000 with an increment of 50. Since the average of calculated F1-scores was highest at the $K^2$ test statistic cutoff of 600 (S1C Fig), this cutoff was used to determine the iModulons [26]. The iModulons containing less than 4 genes were discarded and thus, 72 iModulons were identified eventually.

## Calculation of CEV of PCA and ICA

PCA of the input data set was conducted with the Scikit-learn Python package (v0.22.1) [42]. The CEV of the PCA results was calculated by adding the explained variance ratios of the principal components in the order of numbers using the NumPy Python package [47]. The CEV of the ICA results was calculated by adding the explained variance ratios of the independent components as described above. The Matplotlib Python package was used to visualize the CEV of the PCA and ICA results [48].

## Correlation analysis of the expression levels of the genes or the activities of the iModulons

The expression levels of the genes and activities of the iModulons for each RNA-seq condition were obtained from S2 and S4 Datasets, respectively, by using Pandas and NumPy Python packages [47,49]. The Pearson correlation analyses between the expression levels of a gene and the activities of an iModulon, or between the expression levels of two genes were conducted with the SciPy Python package [50]. The Matplotlib Python package was used to visualize the plots of the correlation analysis [48].

## Prediction, modeling, and visualization of the protein structures

The repeats and domains of the proteins were predicted by using InterPro (v88.0) [51]. The structural predictions of the full-length proteins were obtained from the Alphafold2 algorithm [52]. Visualization of the predicted protein structures and superimposition of two proteins was conducted with PyMOL (v2.5.2) [53].

## Strains, plasmids, and culture conditions

The strains and plasmids used in this study are listed in S5 Table. Unless otherwise noted, the *V. vulnificus* strains were grown aerobically in Luria-Bertani (LB) medium supplemented with 2% (w/v) NaCl (LBS) at 30°C. The *Vibrio fischeri* minimal medium containing glycerol (50 mM Tris-HCl, pH 7.2, 50 mM MgSO$_4$, 300 mM NaCl, 10 mM KCl, 0.33 mM K$_2$HPO$_4$, 18.5 mM NH$_4$Cl, 10 mM CaCl$_2$, and 32.6 mM glycerol) (VFMG) was used for biofilm formation [11]. When required, antibiotics were added to the media at the following concentrations: kanamycin, 100 μg/ml and ampicillin, 100 μg/ml. To manipulate the intracellular c-di-GMP

levels, *V. vulnificus* JN111, which carries *dcpA* encoding a diguanylate cyclase [54] on the chromosome under the control of the arabinose-inducible promoter $P_{BAD}$ [55], was constructed from the wild-type strain *V. vulnificus* CMCP6 previously [14]. JN111 was used as a parent strain in this study (S5 Table), and intracellular c-di-GMP levels of the *V. vulnificus* strains were manipulated by adding different concentrations of arabinose to the growth media.

## Generation and complementation of the deletion mutants

For the construction of the isogenic deletion mutants, target genes were inactivated *in vitro* by deletion of each ORF using the polymerase chain reaction (PCR)-mediated linker-scanning mutation as described previously [56]. Briefly, the deleted ORF fragment was amplified by PCR with appropriate primer pairs (S6 Table), and the amplified fragment was cloned into SphI-SpeI-digested pDM4 [57]. *E. coli* S17–1 $\lambda pir$, containing pDM4 with the desired insert, was used as a conjugal donor to the parent strain to generate the deletion mutant (S5 Table). The conjugation and isolation of the transconjugants were conducted using a method described previously [58]. The *brpN brpL* double mutant was generated identically from the *brpL* mutant by additional deletion of *brpN*. For complementation of the deletions, respective ORF was amplified by PCR with appropriate primer pairs (S6 Table), and the amplified fragment was cloned into NcoI-SphI-digested pJK1113 [59] under an arabinose-inducible promoter $P_{BAD}$. The plasmids were transferred into the appropriate mutants by conjugation as described previously [60].

## RNA purification and qRT-PCR

Relative transcript levels in the total RNA isolated from the *V. vulnificus* strains grown to the stationary phase (an $A_{600}$ of 2.0) in LB (added arabinose and antibiotics when required) were determined by quantitative reverse transcription-PCR (qRT-PCR). Total RNA from the *V. vulnificus* cells was isolated by using an RNeasy mini kit (Qiagen, Valencia, CA, United States) and quantified using a NanoDrop One spectrophotometer (Thermo Scientific, Waltham, MA, United States). cDNA was synthesized from 400 ng of the total RNA using a PrimeScript RT Master Mix reagent kit (Takara, Osaka, Japan) in a final volume of 20 μl. Real-time PCR amplification of the cDNA was performed with the CFX96 real-time PCR detection system (Bio-Rad, Hercules, CA, United States) with appropriate primer pairs (S6 Table) as described previously [61]. Relative expression levels of the target gene were calculated by using the 16S ribosomal RNA, *rrsH*, as an internal reference for normalization.

## Protein purification and EMSA

To overexpress BrpT, pSH1819 carrying the *brpT* gene on pET-28a(+) (Novagen, Madison, WI, United States) was used as described previously [13]. The His6-tagged BrpT was expressed in *E. coli* BL21 (DE3) and purified by affinity chromatography using Ni-NTA agarose (Qiagen). For EMSA, the 279-bp *cabH* upstream region was amplified by PCR using unlabeled CABHUP_F and 6-carboxyfluorescein (6-FAM)-labeled CABHUP_R as primers (S6 Table). Similarly, the 300-bp *brpN* upstream region was amplified by PCR using unlabeled BRPNUP_F and 6-FAM-labeled BRPNUP_R as primers (S6 Table). The 6-FAM-labeled DNA (10 nM) was incubated with different amounts of purified BrpT for 45 min at 25°C in a 20-μl reaction mixture containing 1 × BrpT-binding buffer (10 mM Tris-HCl, pH 7.5, 50 mM KCl, 75 mM NaCl, 10 mM MgCl₂, 1 mM DTT, 0.1 mM EDTA, 1 μg bovine serum albumin, 5% (w/v) glycerol), and 0.1 μg of poly(dI-dC) as a non-specific competitor. For the competition analysis, various concentrations of unlabeled DNA fragments were added as a self-competitor to the

reaction mixture before incubation. Electrophoretic analysis for the DNA-protein complexes was performed as described previously [62].

## Quantitative analysis and visualization of the biofilms

To quantify the biofilms of the *V. vulnificus* strains, each well of the 96-well polystyrene microtiter plates (Nunc, Roskilde, Denmark) was inoculated with 200 μl of culture diluted to an $A_{600}$ of 0.05 in VFMG supplemented with 0.01% (w/v) arabinose (added antibiotics when required). After static incubation for 24 h at 30˚C, supernatants were removed from the wells and the remaining biofilms were stained with 1% (w/v) crystal violet solution for 15 min. The biofilms were quantified by elution of stained biofilms with ethanol and measurement of absorbance at 570 nm ($A_{570}$) as described previously [63]. To visualize the biofilms, biofilms of the strains were formed and stained as described above but in a larger scale (1 ml) using polystyrene (SPL, Seoul, Republic of Korea) or glass round-bottom test tubes. To remove loosely attached cells, the remaining biofilms were washed with a vibration of 1,200 rpm for 20 sec in 1 ml of phosphate-buffered saline (PBS). The stained biofilms were photographed by a mobile camera after the removal of the crystal violet solution or PBS. Additionally, after being washed with PBS, the biofilms were quantified by elution of stained biofilms with ethanol and measurement of $A_{570}$ as described above.

## Colony morphology assay

For the analysis of the colony morphology, 2 μl of cultures grown to an $A_{600}$ of 0.8 were spotted onto VFMG agar supplemented with 0.02% (w/v) arabinose (added antibiotics when required). The resulting colonies grown at 30˚C for 24 h were visualized using a Stemi 305 stereomicroscope (Zeiss, Oberkochen, Germany) equipped with an Axiocam 105 color camera (Zeiss).

## EPS analysis

Exopolysaccharide was prepared following the procedures previously described [64]. Briefly, each culture grown on an LBS agar plate containing 0.02% (w/v) arabinose was suspended in PBS and diluted to an $A_{600}$ of 1.0. The suspensions were vigorously shaken to elute the EPS from the cells. The cells and debris were removed by centrifugation, and the supernatant was treated with RNase A (50 μl/ml), DNase I (50 μg/ml with 10 mM $MgCl_2$), and proteinase K (200 μg/ml). Subsequently, the remaining polysaccharide fraction was extracted with phenol-chloroform, precipitated with $2.5 \times$ volumes of 100% ethanol, and resuspended in distilled water. The EPS resuspensions were resolved on a 4% polyacrylamide gel by SDS-PAGE and stained with Stains-All (Sigma-Aldrich, St. Louis, MO, United States). The gel was subsequently destained as described previously [65] and photographed by a mobile camera. The intensity of stained EPS in each lane was determined using ImageJ software (NIH, Bethesda, MD, United States), and the ratio of the intensity of mutants to that of the parent strain was calculated.

## Data analysis

Average and standard deviation (SD) values were calculated from at least three independent experiments. The experimental data were analyzed by Student's *t* tests using GraphPad Prism 7.0 (GraphPad Software, San Diego, CA, United States). The significance of the differences between experimental groups was accepted at a *P* value of $< 0.05$.

## Supporting information

**S1 Fig. Summary of data processing.** (A) Cumulative explained variance (CEV) for the transcriptome data of *V. vulnificus* was calculated by using principal component analysis (PCA, blue line) and independent component analysis (ICA, orange line). Using the 83 independent components, ICA reconstructed 80% of the total explained variance of the transcriptome data calculated by PCA. (B) A histogram of the gene coefficients in the entire independent components. Most of the gene coefficients were distributed around zero, and the gene coefficients in any independent component display a similar distribution. (C) Average F1-scores calculated under the varied D'Agostino $K^2$ statistic cutoff ranging from 200 to 1,000 with an increment of 50. The optimal cutoff value was identified as 600 where the highest average F1-score was observed (shown in a red dot).
(TIF)

**S2 Fig. iModulons identified by genomic differences between strains.** (A, B) Box plots of the iModulon activities under various conditions of the *V. vulnificus* cells. Box plot whiskers, box bounds, and the center line represent extrema, upper and lower quartiles, and the median value, respectively. MO6-24/O; CMCP6; Δ*nsrR*; Δ*smcR*; Δ*hlyU*; others, transcriptome data from *V. vulnificus* MO6-24/O; *V. vulnificus* CMCP6; the *nsrR* mutant; the *smcR* mutant; the *hlyU* mutant; all other strains except the corresponding mutants, respectively.
(TIF)

**S3 Fig. Identification of the iModulons specifically activated in biofilm cells.** (A to C) Box plots of the iModulon activities under various conditions of biofilm and planktonic cells. Box plot whiskers, box bounds, and the center line represent extrema, upper and lower quartiles, and the median value, respectively. The gray dashed line represents zero. Early and mature biofilm cells indicate the biofilm cells incubated in VFMG for 1.5 h and 6~13 h, respectively. Among the various conditions of planktonic cells in the transcriptome data, conditions using M9G, LBS, and MHB as the growth medium were selected as the representatives. M9G, M9 minimal medium supplemented with 0.4% (w/v) glucose; LBS, Luria-Bertani (LB) medium supplemented with 2% (w/v) NaCl; MHB, Mueller-Hinton broth.
(TIF)

**S4 Fig. Validation of iModulon-40 and iModulon-64.** (A, B) Ordered correlation matrix. Colors indicate the Pearson *R* values between the activities of iModulon-40 (A) or iModulon-64 (B) and the expression levels of the element genes, respectively. Red and blue represent the strongest positive (+1) and negative (-1) correlation, respectively.
(TIF)

**S5 Fig. Differentially expressed genes in the *V. vulnificus* biofilm cells.** (A to C) Volcano plots from differential expression analysis between biofilm and planktonic cells. Total RNAs were isolated from the biofilm and planktonic *V. vulnificus* cells after incubation for 1.5 h (A), 6 h (B), and 10 h (C) in VFMG. Transcriptome analysis plotted the genes down-regulated or up-regulated in the biofilm cells (see S1 Methods for details). The black dots represent differentially expressed genes and the red dots represent *cabH* and *brpN* as indicated. The gray dashed lines indicate the cutoffs for differential expression of fold change > 2 and *P* value < 0.05.
(TIF)

**S6 Fig. Biofilm formation and colony morphology of the *brpN*, *brpL*, and *brpN brpL* double mutants.** (A) Effects of *brpN* and *brpL* deletions on biofilm formation. For quantitative analysis of the biofilm, biofilms of the *V. vulnificus* strains were grown in VFMG supplemented

with 0.01% arabinose in 96-well polystyrene microtiter plates for 24 h. Then supernatants were removed from the wells and the remaining biofilms were stained with 1% crystal violet. The crystal violet was eluted and its $A_{570}$ was determined to quantify the biofilms ($n$ = 3). Error bars represent the SD. Statistical significance was determined by the Student's $t$ test (*, $P$ < 0.05; **, $P$ < 0.005). (B) Effects of *brpN* and *brpL* deletions on colony morphology. The parent strain and mutants were spotted onto VFMG agar supplemented with 0.02% arabinose and incubated for 24 h. Each colony representing the mean rugosity from at least three independent experiments was visualized using a stereomicroscope (Stemi 305, Zeiss). All images are shown at the same scale, and a 1-mm scale bar is shown on the image of the parent strain. Parent, parent strain; Δ*brpN*, *brpN* mutant; Δ*brpL*, *brpL* mutant; Δ*brpN* Δ*brpL*, *brpN* *brpL* double mutant. (TIF)

**S1 Table. Overview of the iModulons identified by ICA.**
(DOCX)

**S2 Table. The element genes included in the iModulon-40.**
(DOCX)

**S3 Table. The element genes included in the iModulon-64.**
(DOCX)

**S4 Table. The GC contents of *brpN*, *brpLG*, the *brp* locus, and chromosome II.**
(DOCX)

**S5 Table. Bacterial strains and plasmids used in this study.**
(DOCX)

**S6 Table. Oligonucleotides used in this study.**
(DOCX)

**S1 Dataset. Information of raw sequencing reads of *V. vulnificus* transcriptome data retrieved from the SRA database at NCBI.**
(XLSX)

**S2 Dataset. Input data set of ICA comprising 4,264 genes and 161 conditions.**
(XLSX)

**S3 Dataset. Gene coefficients of 4,264 genes constituting the 83 independent components.**
(XLSX)

**S4 Dataset. Activities of the 83 independent components along the 161 conditions.**
(XLSX)

**S5 Dataset. Identified iModulons and their element genes.**
(XLSX)

**S1 Data. Underlying numerical values of Figs 5A, 5B, 6A, 6B, 8C, 8D, and 9B and S1A, S1C, S5A, S5B, S5C, and S6A Figs.**
(XLSX)

**S1 Methods. Differential expression analysis of the genes in biofilm and planktonic cells.**
(DOCX)

## Author Contributions

**Conceptualization:** Hojun Lee, Sang Ho Choi.

**Data curation:** Hojun Lee, Hanhyeok Im, Seung-Ho Hwang.

**Formal analysis:** Hojun Lee, Hanhyeok Im, Seung-Ho Hwang.

**Funding acquisition:** Hojun Lee, Sang Ho Choi.

**Investigation:** Hojun Lee, Hanhyeok Im, Seung-Ho Hwang, Duhyun Ko.

**Methodology:** Hojun Lee, Hanhyeok Im, Duhyun Ko.

**Software:** Hojun Lee, Hanhyeok Im.

**Supervision:** Sang Ho Choi.

**Validation:** Sang Ho Choi.

**Visualization:** Hojun Lee.

**Writing – original draft:** Hojun Lee, Hanhyeok Im, Duhyun Ko, Sang Ho Choi.

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
