## [Decision Letter · Decision Letter 0]

1 Nov 2022

Dear Dr. Choi,

Thank you very much for submitting your manuscript "Two novel genes identified by large-scale transcriptomic analysis are essential for biofilm and rugose colony development of Vibrio vulnificus" for consideration at PLOS Pathogens. As with all papers reviewed by the journal, your manuscript was reviewed by members of the editorial board and by several independent reviewers. The reviewers appreciated the attention to an important topic. Based on the reviews, we are likely to accept this manuscript for publication, providing that you modify the manuscript according to the review recommendations.

Sincerely,

Matthew Parsek, PhD

Associate Editor

PLOS Pathogens

Karla Satchell

Section Editor

PLOS Pathogens

Kasturi Haldar

Editor-in-Chief

PLOS Pathogens

orcid.org/0000-0001-5065-158X

Michael Malim

Editor-in-Chief

PLOS Pathogens

orcid.org/0000-0002-7699-2064

Reviewer Comments (if any, and for reference):

Reviewer's Responses to Questions

**Part I - Summary**

Reviewer #1: In this manuscript, Lee and colleagues use a machine learning algorithm to identify groups of co-regulated genes, termed iModulons, in the Vibrio vulnificus genome linked to biofilm formation. The bioinformatic deep-dive was executed by using independent component analysis (ICA) to re-analyze hundreds of transcriptomic datasets publicly available through NCBI. This approach was validated by identifying 2 additional genes, cabH and bprN, that are part of the BpsT regulon, which regulates extracellular polysaccharide production and biofilm formation in V. vulnificus.

The manuscript is technically sound. The approach to genetic analysis for cabH and bprN is thorough – including phenotyping of mutants, complementation analysis, and demonstration of BpsT-dependent gene expression using EMSAs, which are not trivial. Taken together, this comprehensive approach provides confidence in the genetic linkages and provides proof-of-principle for the bioinformatic approach. I enjoyed reading this article.

Reviewer #2: The ability to form multicellular bacterial communities, or biofilms, is predominant among bacterial species. Biofilm formation provides protection to the bacterial community and allows opportunities for nutrient acquisition/concentration and horizontal gene transfer. This protected communal lifestyle allows for the survival and persistence of bacteria in nature. Among biofilm-forming bacteria, are naturally occurring aquatic Vibrio species. Vibrionaceae species such as V. cholerae, V. parahaemolyticus, and V. vulnificus are facultative human pathogens, with significant and devastating impacts to public health. Biofilm formation is key to the environmental survival and persistence of Vibrio species and can impart a substantial role in host pathogenesis and disease. In this work, Lee et al. performed an in-depth analysis of transcriptome data comparing planktonic and biofilm modalities within the seafood-borne pathogen V. vulnificus. The objective was to use an unbiased machine learning algorithm, Independent Component Analysis (ICA), to identify genetic regulons (iModulons) related to biofilm and rugose colony formation in V. vulnificus.

The authors identified several biofilm-associated iModulons; one of which contained genes modulated by the known V. vulnificus biofilm transcriptional regulator, BrpT. Along with genes known to be regulated by BrpT, the authors identify two novel BrpT-regulated genes of unknown function: VV1_3061(cabH) and VV2_1694 (brpN). The authors next validate these two genes as BrpT-regulated through qRT-PCR, and predict their function based on structural homology to known V. vulnificus genes. The authors then beautifully demonstrate that CabH appears to be involved in surface-attachment, and BrpN appears to be involved in production or secretion of exopolysaccharide. Overall, I feel that the quality of work and data analysis is strong, and that the results match the interpretation developed by the authors.

I feel that the work presented here is novel and represents a very significant advancement for the study of biofilm regulation in bacteria. The authors present a very well written and detailed step-by-step procedure for the computational identification of previously unknown genes involved in V. vulnificus biofilm formation, and for molecular verification of results predicted by computational analysis. Not only is this work significant for the study of both V. vulnificus and biofilm biology, but it is also a significant step forward in development and utilization of unbiased data analysis methods for the interpretation of large transcriptome datasets. I do feel that this work is acceptable for publication in PLoS Pathogens and would be of key interest to its audience. Therefore, with consideration of the minor revisions outlined below, I would recommend publication of this work.

**Part II – Major Issues: Key Experiments Required for Acceptance**

Reviewer #1: 1. Perhaps the most interesting thing in the paper is the analysis of transcriptional regulatory networks in V. vulnificus using ICA to identify iModulons. However, relatively little information about this machine learning approach is presented in the manuscript. The descriptions leave me wanting to know more. I suspect it will be the same for other readers. Could the authors provide an expanded description of these methods and the “Modulome” workflow? This is exciting, new technology.

2. There is a lot of information yielded by the Modulome workflow that is glossed over in the manuscript. Could the authors consider providing a display item that summarizes some of the insights into V. vulnificus regulatory networks provided by this analysis? I realize that this is not a small ask – but it comes across as an omission. As a microbiologist, I want to know more about what machine learning can reveal about gene expression in this bacterium.

3. Can the authors add their datasets to iModulonDB? See (https://academic.oup.com/nar/article/49/D1/D112/5921295).

Reviewer #2: (No Response)

**Part III – Minor Issues: Editorial and Data Presentation Modifications**

Reviewer #1: 1. In all bar graphs, could the authors please show their individual datum points so that the readers can evaluate data distributions for themselves.

2. While it is possible that rendering in Editorial Manager may have affected figure quality, it is very difficult to see rugose colony morphology in the figures. Could the authors double check the image quality, or alternatively, take better photos so that rugosity is easily visible to the reader.

Reviewer #2: Minor Modifications:

- Figure 3A-B: How many biological replicates were used for qRT-PCR analysis? Please state in the Figure Legend or Materials and Methods.

- Figure 3C-D: How many independent times were the EMSA analyses performed? Provided gel images are representative of how many assays? Please state in the Figure Legend or Materials and Methods.

- Figure 4: How many biological replicates were used for the crystal violet staining biofilm assay? Please state in the Figure Legend or Materials and Methods.

- Figure 6: How many biological replicates were used for the crystal violet staining biofilm assay? Please state in the Figure Legend or Materials and Methods.

- Figure 6: While I agree with the conclusions of the authors that CabH appears to be involved in surface attachment, and BrpN impacts biofilm formation independent of attachment, I do feel that this assertion could be strengthened by quantification of the crystal violet used for staining here after washing of the biofilm. This would not only allow for quantification of the differences between the isogenic parent strain and each mutant tested but would also allow for determination of more subtle differences between the strains.

This assertion could also be strengthened by the microscopic analysis of the ability of the cabH mutant to attach to surfaces. Is the mutant impaired in initial attachment, or can it initially attach to a surface and the impairment is in the long-term ability to stay on a surface (given that the biofilms are analyzed at the mature state od 24H)? While this analysis may not be required for this publication, I do believe it is something that the authors should consider.

PLOS authors have the option to publish the peer review history of their article (what does this mean?). If published, this will include your full peer review and any attached files.

Reviewer #1: **Yes: **Joe J. Harrison

Reviewer #2: **Yes: **Kyle A. Floyd

Figure Files:

Data Requirements:

Reproducibility:

References:

---

## [Editor Report · Decision Letter 1]

13 Dec 2022

Dear Dr. Choi,

We are pleased to inform you that your manuscript 'Two novel genes identified by large-scale transcriptomic analysis are essential for biofilm and rugose colony development of Vibrio vulnificus' has been provisionally accepted for publication in PLOS Pathogens.

Best regards,

Matthew Parsek, PhD

Academic Editor

PLOS Pathogens

Karla Satchell

Section Editor

PLOS Pathogens

Kasturi Haldar

Editor-in-Chief

PLOS Pathogens

orcid.org/0000-0001-5065-158X

Michael Malim

Editor-in-Chief

PLOS Pathogens

orcid.org/0000-0002-7699-2064
---

## [Editor Report · Acceptance letter]

13 Jan 2023

Dear Dr. Choi,

We are delighted to inform you that your manuscript, "Two novel genes identified by large-scale transcriptomic analysis are essential for biofilm and rugose colony development of Vibrio vulnificus," has been formally accepted for publication in PLOS Pathogens.

Best regards,

Kasturi Haldar

Editor-in-Chief

PLOS Pathogens

orcid.org/0000-0001-5065-158X

Michael Malim

Editor-in-Chief

PLOS Pathogens

orcid.org/0000-0002-7699-2064